# Assessment of Berries of Some Sea Buckthorn Genotypes by Physicochemical Properties and Fatty Acid Content of the Seed

**DOI:** 10.3390/plants11243412

**Published:** 2022-12-07

**Authors:** Mónika Máté, Granit Selimaj, Gergely Simon, Lilla Szalóki-Dorkó, Gitta Ficzek

**Affiliations:** 1Department of Fruit and Vegetable Processing Technology, Institute of Food Science and Technology, Hungarian University of Agriculture and Life Sciences, 1118 Budapest, Hungary; 2Department of Fruit Growing, Institute of Horticulture, Hungarian University of Agriculture and Life Sciences, 1118 Budapest, Hungary

**Keywords:** sea buckthorn (*Hippophae rhamnoides* L.) cultivar, candidate, physical parameters, color coordinates, fatty acids, processability, GC-FID

## Abstract

Sea buckthorn (*Hippophae rhamnoides* L.) is consumed mainly in its processed form. Therefore, the investigation of the physicochemical properties of its berries is a current task in the aspect of food processing. The aim of this study was to determine the physicochemical parameters (soluble solid content, total titratable acidity, sugar/acid ratio), color characteristics (L*, a*, b*) and fatty acid profile of five varieties (‘Askola’, ‘Clara’, ‘Habego’, ‘Leikora’, ‘Mara’) and one Hungarian candidate, R-01, to establish a basis for experiments on the processability of the whole berries (e.g., drying). The weight of the berry of ‘Leikora’ (0.64 g) was significantly higher than the other investigated fruits. The differences between the values of soluble solid content (6.3–10.84 °Brix) and titratable acid (1.4–3.7%) content of berries were significant. ‘Mara’ had the highest sugar/acid ratio. Regarding the fatty acid profile, the amount of unsaturated fatty acids was measured between 72.6–83.4%, including polyunsaturated fatty acids, which were between 32.3–58.1%. The seeds of the tested samples contained high concentrations of linoleic acid (17.0–33.2%) and linolenic acid (15.3–24.9%), mainly in the case of the ‘Mara’, ‘Clara’ and ‘Askola’ varieties. Candidate R-01 could be used as a raw material for functional foods due to its significant content of palmitoleic acid and a favourable omega-6/omega-3 ratio.

## 1. Introduction

Sea buckthorn (*Hippophae rhamnoides* L.) is a widely cultivated plant in Europe, Asia and Canada. Its unique value is due to its large amounts of both water-soluble (e.g., ascorbic acid, phenolic acids, flavonoids, tannins) and fat-soluble (e.g., carotenoids, tocopherols) antioxidants. Sea buckthorn (SB) berries consist of pulp (55–70% by weight), seeds (10–23% by weight) and peel (5–9% by weight), which are typically rich in oil. However, the proportions are very different due to the genetic diversity of SB, which are also important properties for the processing industry.

Several scientific findings have been published on the high concentration of bioactive compounds in SB berries in connection with their in vivo and in vitro health benefits [1,2,3,4,5]. However, the effects of processing and storage on these compounds are less well known.

Due to the bitter, astringent taste and high acid content, the fresh consumption of SB berries is not preferred. It is most commonly consumed in the form of fruit juice and puree, but it is usually mixed with other fruits that soften/mitigate the sour taste or sweeten it [6].

To preserve the biologically active ingredients, the processing industry prefers modern, gentle preservation methods, e.g., aseptic treatment and filling or high-pressure treatment. In addition to the liquid form, the preservation of berries as a whole is becoming more and more popular, which primarily means dehydration technology, e.g., the lyophilization, vacuum or microwave vacuum drying of whole berries. The processing technology affects the quantity and transformation of the active ingredients of the berries and has effects on the subsequent storability of the product. The suitability of berries for drying (e.g., rate of moisture release, color changes, shape change, puffability, etc.) is greatly influenced by the size of the berries, the seed/berry ratio, the dry matter content and the color. The sensory properties of dried products are significantly determined by the amount and ratio of sugars and acids [7]. Cultivars with a higher sugar/acid ratio are more prevalent in the processed products [8]. Using whole SB berries in different products, e.g., mixing dried berries into teas, muesli mixes, slices, and snack products, increases their biological value due to the high content of ascorbic acid, carotenoids, phenolic substances and fatty acids. However, the presence of fatty acids also affects the shelf life of the products.

The specialty of SB is that the whole fruit contains fatty acids. The oil content of the seeds is 100–200 g kg^−1^ [9,10,11], while for the flesh and shell parts it is between 20–105 g kg^−1^ in the fresh state. SB pulp oil contains approximately 48% saturated and 52% unsaturated fatty acids, while the seed oil contains 12–20% saturated and 80–90% unsaturated fatty acids. The fatty acid profile is essential for evaluating the nutritional value of berries, especially the ratio of polyunsaturated fats.

Nineteen fatty acids have been identified from SB berries, of which eight are saturated and eleven are unsaturated fatty acids. The ratio of saturated fatty acids is between 13.70–42.68%, while monounsaturated fatty acids can be detected between 40.73–60.37% and polyunsaturated fatty acids between 3.70–24.62%. Among the essential fatty acids, linolenic acid and linoleic acid are significant [12,13]. Oils from SB seeds and pulp have different fatty acid compositions. Palmitic acid, palmitoleic acid and oleic acid is considerably higher in the flesh, while linoleic acid, linolenic acid and oleic acid predominate in the seed [14].

Sea buckthorn seed oil contains 7–44% linoleic acid, 27–31% linolenic acid and 17–20% oleic acid. The ratio of omega-6/omega-3 (linoleic acid/linolenic acid) polyunsaturated fatty acids is usually 1:1 [15]. Additionally, SB oil is the only oil that naturally contains a 1:1 omega-3:omega-6 ratio [4]. The saturated fatty acid content of the seed oil is approximately 10–13%, with the dominant saturated fatty acids being palmitic acid (7–9%) and stearic acid (2.5–3%) [9,10,16].

Fatty acids play a fundamental role in human health. Many studies have reported on the clinical use of SB oil. The composition of fatty acids found in SB oil is unique and provides many health benefits for the human body, which is why it is highly valued by both the biomedicine and cosmetic industries. They are effective against inflammatory processes, protect epithelial tissues and have a beneficial effect on the digestive organs, respiratory organs, urological organs, female genital organs and the inside of the eye (dry eye syndrome) [17,18]. Chand et al. [19] found that the quality and cholesterol level of egg were significantly improved when laying hens were fed 2 and 3 g/kg SB supplements.

The genetic diversity of SB is enormous, and, accordingly, there can be significant differences in the biologically active ingredient content of individual varieties. However, there are only few data available about the biologically active ingredient content of grown cultivars. To obtain more information about the health-protective value of recent varieties and to clarify the possibilities of their use, a substantial analysis of the individual genotypes is highly important.

The aim of this study was to determine the physical parameters, color characteristics and fatty acid profile of five cultivars and one promising candidate to establish and support the processing of whole berries, e.g., with drying and their storage.

## 2. Results

### 2.1. Physical Properties of Berries of the Different SB Cultivars

The raw fruits of SB cultivars are not consumed, so the price of the cultivars is less influenced by the size parameters. However, in the case of some processing industry aims, the size plays an important role, e.g., the suitability of the whole berry for drying. The difference between the size parameters (height, thickness, width) of the berries of the investigated cultivars were significant (Table 1.). The highest value was represented in the case of the fruit of ‘Leikora’, followed by ‘Clara’, ‘Habego’ and the candidate R-01. However, in the case of the latter-mentioned cultivars, the significant difference in size parameters in the berry weight could no longer be verified, which can probably be explained by the different genetically determined berry shape, flesh firmness, density and texture. The weight of the berry of ‘Leikora’ (0.64 g) was statistically greater than the fruit of the other tested cultivars. In the case of some processing industry purposes (e.g., extracting juice and pulp production, drying whole berries), in addition to the size of the berries, the seed weight and the seed/berry ratio also play an important role. The largest seed weight was the fruit of ‘Habego’ which was followed by the candidate R-01, ‘Leikora’ and ’Ascola’, although no significant difference in the seed/berry ratio of the berries of the cultivars could be observed. So, in the case of the tested cultivars, the larger berry mass was in correlation to a proportionally larger seed mass.

### 2.2. The Soluble Solid Content and Total Titratable Acid Content of the Berries of SB Cultivars

The soluble solid content (SSC) and total titratable acidity (TTA) of the fruits, as well as their relative ratio, have a special role from the point of view of fresh consumption and the processing industry. Based on the observation of Tiitinen et al. [20], in the case of SB, the soluble solid content and the sugar/acid ratio were positively correlated with sweetness and negatively associated with sourness and astringency. Therefore, their examination is vital from the aspect of the sensory properties of the cultivars, as well as their consumer perception and acceptance. Previous research results prove that there are significant differences in the soluble solid content of SB genotypes (2.9–35.2 °Brix) [21,22,23,24]. The soluble solid content of the SB cultivars was between 6.3 and 10.9 °Brix (Table 2) in our present research. The berries of the candidate R-01 (10.25 °Brix) and ‘Mara’ (10.9 °Brix) had a statistically verifiable higher SSC compared to the other examined cultivars. Our results were similar to those studied in our previous research, where the fruits were also grown under ecological conditions in Hungary, but the SSC of the berries of ‘Leikora’ (5.6 °Brix) and ‘Ascola’ (7.9 °Brix) were grown from a different location (46°25′05″ N, 18°55′08″ E) [23]. Mezey et al. [25] measured 4.81 °Brix in the case of the ‘Leikora’ cultivar in the south-west Slovakian growing area, which was lower than our measured values. Furthermore, significantly higher values were measured in Turkish (10.1–14.8 °Brix) [26] and Iranian wild genotypes (8.6–35.2 °Brix) [22]. Green et al. [21] measured between 8.9–12.5 °Brix in varieties grown in Canada, while in Finland it was much lower with a value of 1.9–7.1 g/100 g [20]. The phytochemical and nutritional composition of SB berries depends on the cultivars, climatic and growing conditions, the year effect, ripening status, storage conditions, harvesting time and the method of processing and analysis [26,27]. In addition, the sugar content varies depending on the origin, population and genetic background of the plant [22].

Based on our results, fruits of the R-01 candidate had the highest TTA (3.7 g 100 g^−1^), followed by ‘Ascola’ (2.7 g 100 g^−1^) and ‘Habego’ (2.3 g 100 g^−1^), while the fruit of the other observed cultivars showed lower but statistically identical values. Ercisli et al. [28] measured 2.64–4.54 g 100 g^−1^ total titratable acidity values in Turkish wild species, while Tang and Tigersted [29] detected slightly higher (3.25–4.46 g 100 g^−1^) values in a hybrid population. In the case of Finnish cultivated cultivars, it was significantly higher at 3.1–5.1 g 100 g^−1^ [20]. In the case of cultivars grown in Canada, the acid content was significantly lower, with values between 1.42–1.89 g 100 g^−1^ [21]. Mezey et al. [25] measured a higher acid content of 2.61 g/100 g and a significantly lower Brix degree (4.81 °Brix) than the value measured by us (2 mg 100 g^−1^) in the case of the ‘Leikora’ cultivars. So, the sugar/acid ratio (1.81) of the ‘Leikora’ cultivar grown in the southwestern Slovak growing area was significantly lower than the results of the present study.

The fruit of ‘Mara’ had the highest SSC content and the lowest acid content among the studied cultivars, resulting in an outstandingly high sugar/acid ratio (7.8). Although the °Brix value and titratable acid content of the examined genotypes were significantly different, their sugar/acid ratio was statistically verifiably the same, except for ‘Mara’ (2.7–3.9). Ma et al. [24] measured 7–11 °Brix and a 4.0–9.3 g/100 g acid content in the case of cultivars grown in Estonia, so the sugar/acid ratio was between 0.3–1.3, which is a much lower value compared to the domestically grown cultivars.

### 2.3. Color Characteristics of the Berries of SB Cultivars

From the point of view of the processing industry, the color of the raw material is of outstanding importance because it affects the color and shade of the final product made from it as well as the color stability during processing and storage. The color of SB berries ranges from yellow to orange-red, sometimes with red hues, so the color variability is very high [16]. The CIELab color stimulus coordinates are suitable for characterizing the color and showing the differences between cultivars. There were significant differences in the color of the fruit of the SB cultivars for all five investigated parameters (Table 3). The yellow color component and, in connection with this, the color saturation value and the hue in the fruits of ‘Clara’ and ‘Habego’ showed high values. Based on the measured a*, b* coordinates, the cultivar ‘Ascola’ could be characterized by the lowest color saturation value (Chroma) and hue angle (Hue). The reason for the differences in the measured color parameters was probably due to the concentration and accumulation of carotenoids. Our previous research proved that there are significant differences in the carotenoid content of the fruits of each genotype [23,30].

Green et al. [21] measured a lightness factor (L*) of 46.1–48.5 in the case of cultivars grown in Canada, which is most similar to ‘Mara’ and ‘Ascola’ among the domestically grown cultivars. Regarding the red–green ratio (a*), the cultivars showed values between 17.3 and 33.8, which indicated that they were more reddish. Among the domestically grown cultivars, ‘Clara’, ‘Habego’, ‘Leikora’ and ‘Mara’ showed similarities. Regarding the yellow–blue (b*) ratio, values between 37.3 and 59.1 were measured, and among the investigated cultivars in this study, ‘Clara’, ‘Habego’ and ‘Leikora’ were similar. Chroma refers to the strength or intensity of the color, i.e., the saturation of the color. According to Green et al. [21], in the tested cultivars, the value of the cultivars was between 60.2–70.5. The color hue values were between 41.8–68.2. These were similar to our investigated domestically grown cultivars. However, it is not enough to know the color parameters of the raw material since the color can change significantly during processing. Therefore, further investigations are necessary to get to know the color change in the individual genotypes due to technological changes.

### 2.4. Fatty Acid Composition of SB Seeds

The results of the fatty acid composition of the SB seeds are shown in Table 4 and Table 5. Chromatograms of the candidate R-01 and a detailed list of the analytes, trivial names, retention times and resolution are shown in the Appendix A.

The content of saturated fatty acids ranged from 16.06 to 27.4%. The seed of ‘Mara’ had the lowest content of saturated fatty acids (16.6%), while candidate R-01 had the highest (27.4%). These values were lower than those in the case of the Turkish cultivars (31.4%) that were measured by Cakir [31].

The total amount of unsaturated fatty acids was between 72.6–83.4%, including polyunsaturated fatty acids between 32.3–58.1% (Table 5.). Cakir [31] determined a value of 68.6% for the unsaturated fat content in the case of Turkish cultivars. Among the polyunsaturated fats, linoleic acid (17.0–33.2%) and linolenic acid (15.3–24.9%) were found in the seeds of the examined samples. In this regard, the cultivars ‘Mara’ and ‘Askola’ were outstanding. Ciesarova [32] analyzed and compared more than 20 pieces of data from different international studies; the content of SB seed linoleic acid was 35–40%, while the content of linolenic acid was between 20–35%. In our investigated cultivars, the amount of linoleic acid and linolenic acid were a few percent lower compared to this international data.

Linoleic acid is an unsaturated omega-3 fatty acid that is essential and must be taken into the human body with nutrition. They are physiological components of cell membranes and mitochondrial membranes and play a role in the mechanism of cell transport and the transmission of neuronal signals [33]. Its ratio is different between subsp. *rhamnoides* and subsp. *sinensis* subspecies. [9]. Linolenic acid is a polyunsaturated omega-6 fatty acid that the human body cannot synthesize on its own, but it is necessary for the normal growth of children [4].

Several studies have reported that human health is improved on a diet in which the ratio of omega-6 to omega-3 essential fatty acids is approx. 1, while in the Western diet, this ratio is 15/1–16.7/1. Over the past 30 years, the total fat and saturated fat intake as a percentage of total calories has decreased in the Western diet, while omega-6 fatty acid intake has increased, and omega-3 fatty acid intake has decreased. As a result, people of today often follow a diet where the omega-6/omega-3 ratio is 20:1 or even higher. This change in fatty acid composition is proportional to a significant increase in the prevalence of overweight, cardiovascular disease, cancer, inflammatory and autoimmune diseases and obesity [34]. In the secondary prevention of cardiovascular diseases, a 4/1 ratio was associated with a 70% reduction in total mortality. A 2.5/1 ratio reduced the proliferation of cells in colon and rectal cancer patients and reduced the risk factor for breast cancer in women. A lower ratio of omega-6/omega-3 fatty acids is more desirable to reduce the risk of chronic diseases commonly found in Western societies and developing countries [34]. Among the studied cultivars, ‘Habego’ (0.96), ‘Leikora’ (1.05) and R-01 (1.1) were the most favorable in terms of the recommended intake of omega6/omega3, but the ratio of the other cultivars was also max. 1.33.

Among the monounsaturated fatty acids, palmitoleic acid should be highlighted, which was present between 1.5–8% in SB seeds based on a summary study by Ciesarova [32]. All our investigated varieties exceeded these values and were in the range from 8.9–22.3%. The candidate R-01 was particularly highlighted, as it contained three times more palmitoleic acid than ‘Mara’ and approx. 1.5 times more than the other cultivars. Palmitoleic acid (16:1) is a monounsaturated omega-7 fatty acid that is very rare in the plant world, so it is very difficult to get it into the human body through plant food sources. SB is one of the few plants that contain this fatty acid. According to most studies it occurs more in the flesh and skin of the berry and less in the seed [9]. Palmitoleic acid is responsible for the self-disinfecting activity of the skin, improves skin and mucous membrane disorders, and improves the occurrence of vaginal inflammatory atrophy [35].

Oleic acid is a monounsaturated omega-9 fatty acid whose advantage over other monounsaturated fatty acids is that it is resistant to oxidation. Both the seed and flesh parts are rich in oleic acid. The seed contains between 15–26% while the pulp contains 10–26% [36]. Several studies have shown that oleic acid reduces the occurrence of cell adhesion molecules in the monolayer squamous epithelium, and this may be associated with its anti-atherosclerotic effect [37].

PCs are linear combinations of the original variables and are determined so that the first PC explains the largest part of the total variance. This means that correlated variables are explained by the same PC and less correlated variables by a different PC. In the present analysis, the two first PC values explained 39.4 and 27.6%, respectively, of the total variance in the case of 13 variables. The score plot shows the total of 67% of the variance (Figure 1). In the PCA plot with two components, four distinct groups were identifiable. The first group was correlated with the R-01 class, the second group with the ’Clara’, ’Leikora’ and ’Habego’ class, the third group was associated with the ’Ascola’ and the fourth group with ’Mara’. The first principal component showed a high correlation (>0.900) with arachinic acid, palmitoleic acid and oleic acid, while the second principal component was correlated with palmitic acid and gondoic acid based on correlation matrix (Table 6). The data for ‘Ascola’ showed the highest variance among the SB samples.

## 3. Materials and Methods

### 3.1. Plant Material

In this research, we examined the fruits of five SB cultivars, ‘Askola’, ‘Clara’, ‘Habego’, ‘Leikora’, and ‘Mara’, and a candidate, R-01 (Figure 2). The research plant materials came from the organic plantation of Superberry Plus Ltd. from Rákóczifalva (47°11′87″ N, 20°21′97″ E), Hungary, in the year 2020. The berries of the studied cultivars were harvested in the full ripening stage (from the beginning of September to the end of October) and after they had reached the characteristic berry color of the cultivar (this parameter was determined visually). Since the peduncle was difficult to separate from the cane, The berries were removed from the pruned fruiting canes one by one with pruning shears. The fruit samples (3 kg/genotypes) were transported to the laboratory immediately after the harvest, where the physicochemical parameters of the samples were determined after arrival. The extracted seeds were stored frozen at −28 °C until determining the fatty acid composition.

### 3.2. Physical and Physicochemical Parameters

The physical parameters of at least 30 berries per sample were examined. The size parameters of the fruits (height, width, thickness) were determined with millimeter accuracy using a digital caliper (Mitutoyo CD-15DC, Mitutoyo Ltd., Telford, UK) that could be connected to a computer. The weight of the tested fruits and the seed weight were measured on a digital scale (KPZ-2-05-4/6000, Klaus-Peter Zander GmbH, Hamburg, Germany). The flesh/seed ratio was calculated from the berry weight and the seed weight (the weight of the seed/the weight of the berry * 100).

The soluble solid content of the homogeneous, filtered fruit juice was determined in °Brix (g 100 g^−1^) with a digital refractometer ((ATAGO Palette PR-10, Atago Co., Ltd., Tokyo, Japan) according to Codex Alimentarius 3-1-558/93 [38]. The titratable acid content was determined in accordance with the Hungarian standard of MSZ EN 12147:1998 [39]. The total acid content (m/m%) was given in malic acid equivalents. The sugar/acid ratio was calculated from the ratio of the water-soluble dry matter content to the titratable acid content.

### 3.3. Determination of Color Coordinates 

The color of the fruits (coordinates L*—lightness, a*—redness, b*—yellowness) was determined with a Konica Minolta tristimulus colorimeter (CR-400, Konica Minolta, Inc., Tokyo, Japan) with 20 color readings for all varieties. To calibrate the instrument, we used the calibrating white tile standard produced by the manufacturer. According to the CIE (Commission Internationale de la Éclargie) standard of 1931, which is also accepted in Hungary, color can be described with coordinates (L*, a*, b*) placed in a 3D color space [40]. L* is the lightness factor in the CIE system; depending on the signs, it characterizes +a* as red, −a* as green, +b* as yellow and −b* as blue. CIELAB uses the color saturation characteristic, where the so-called “chroma” value is interpreted in the a*, b* plane as given in Equation (1), which is the absolute value of the vector, i.e., the distance of the brightness from the axis. The hue angle marked with h indicates the rotation of the direction of the color vector from the direction of the a* axis to the vector C*ab in the color space, so its value can range from 0° to 360° as given in Equation (2). Colors corresponding to hue angle values are red-purple 0°, yellow 90°, bluish-green 180° and blue 270° [41].
C*ab = (a* + b*)^1/2^(1)
h°ab = arctg b*/a*(2)

### 3.4. Determination of Fatty Acid Profile by GC-FID Method

Lyophilized seeds (1 g) of SB cultivars were extracted with petroleum ether (10 mL;15 h) after destruction with 1 M sulfuric acid (5 mL 10%). The fatty acid composition of the fat extracted from the samples was determined in accordance with the ISO 12966-2:2018 standard [42] with minor modifications by Tormási and Abrankó [43]. Between 10–15 mg of fat was measured into a 15 mL centrifuge tube, to which 1.8 mL of isooctane and 200 µL of internal standard (1 mg mL^−1^ glyceryl trinonadecanoate dissolved in chloroform) were added. After dissolving the fat, the esterified fatty acids in the sample were methylated with 200 µL of potassium hydroxide (stirring for 1 min), and after resting (2 min), 4 mL of saturated sodium chloride solution was added to the sample and homogenized (10 s). Separation of the phases was assisted by centrifugation (3700 g, 10 min). Then, the upper (isooctane) phase was transferred to 0.5 g of sodium sulfate salt for dehydration. The sample prepared in this way was analyzed using the GC-FID method. A FAME mixture was used for the qualitative determination of 37 components, and a 4-point calibration (0, 10, 20, 40 µg mL^−1^, supplemented with 100 µg mL^−1^ nonadecanoic acid (dissolved in 1 mg mL^−1^ isooctane)) was used for quantitative determination. 

To determine the fatty acids, an Agilent (Santa Clara, CA, USA) 6890 GC-FID system equipped with an Agilent 7683 autosampler was used. For separation, a Phenomenex (Torrance, CA, USA) Zebron ZB-FAME (60 m, 0.25 mm, 0.20 μm) column with a cyanopropyl stationary phase and a hydrogen gas (1.2 mL/min) mobile phase was used. The inlet temperature was 250 °C, and the detector temperature was 260 °C. A split ratio of 50:1 and a 1 μL injection volume were used. The temperature program started from 100 °C, which was kept constant for 3 min. Then, the column was heated at 20 °C/min to reach 166 °C, where it was kept for 5 min. Then, it was heated to 180 °C at 1 C/min, and finally it was heated to 240 °C at 10 °C/min, where it was kept for 3 min. 

The following fatty acids were determined: myristic acid (C14:0; R_t_ min: 9.94), pentadecanoic acid (C15:0; R_t_ min: 11.05), palmitic acid (C16:0; R_t_ min: 12.43), palmitoleic acid (C16:1n-7c; R_t_ min: 13.24), stearic acid (C18:0; R_t_ min: 16.30), oleic acid (C18:1n-9c; R_t_ min: 17.20), linoleic acid (C18:2n-6c; R_t_ min: 19.04), α-linolenic acid (C18:3n-3c; R_t_ min: 21.56), arachidic acid (C20:0; R_t_ min: 21.59), gondoic acid (C20:1n-9c; R_t_ min: 23.00), dihomo-γ-linolenic acid (C20:3n-6c; R_t_ min: 26.63), behenic acid (C22:0; R_t_ min: 27.91) and cis-15-tetracosenoic acid (C24:1n-9c; R_t_ min: 31.02) (Appendix A).

### 3.5. Statistical Analysis

The data were evaluated using the SPSS 27.0 program (SPSS Inc., Chicago, IL, USA) using the MANOVA test. The separation of homogeneous groups was checked with the univariate Tukey test, and the RSD value was 5% (*n* = 30).

Principal component analysis (PCA) (SPSS software v. 27) was used to compare multiple independent groups in the case of the fatty acid profile. The variance eigenvalue was greater than 1. The loadings (or factor scores) corresponding to the principal components were calculated from the correlation matrix.

## 4. Conclusions

The examination of the physical, biochemical and sensory properties of SB cultivars as well as the effect of processing technologies on individual cultivars requires continuous research. The spread of mild processing procedures provides an opportunity to preserve the bioactive active ingredient as much as possible; therefore, mapping their applicability is of particular importance. Until now, the processing industry has concentrated only on the production of juice, pulp and dietary supplements in the case of SB berries. The great advantage of drying whole berries is that the flesh, skin and seed are processed together. Therefore, these are consumed together as well. However, it is crucial to approach and examine the properties of the berries from this point of view, i.e., the examination of the physical parameters, color characteristics, sugar/acid ratio and fatty acid profile of the berries. The physical parameters determine the processability, the setting of the drying parameters and the sugar/acid ratio determine the sensory properties, and the fatty acid profile characterizes the effect on human health and the rancidity and shelf life of the dried products. Based on the tested parameters, ‘Mara’, ‘Clara’ and ‘Ascola’ were promising cultivars concerning their sugar/acid ratio. Regarding the fatty acid profile, ‘Mara’ also stood out with its high proportion of unsaturated fatty acids. From the point of view of functionality, the candidate R-01 was essential to highlight because of its outstanding palmitoleic acid content and favorable omega 6/omega 3 ratio.

The results indicated that it is worth using the SB seed alone or in the form of a whole berry for the production of functional or enriched foods, especially in their dried form, where the biologically valuable components are present in a concentrated form. They can be used to improve the dietary omega 6/omega 3 ratio, e.g., in bakery products, muesli bars or food supplements. Therefore, in the future it is worth studying the effect of drying methods on the fatty acid composition and their changes.

## Figures and Tables

**Figure 1 plants-11-03412-f001:**
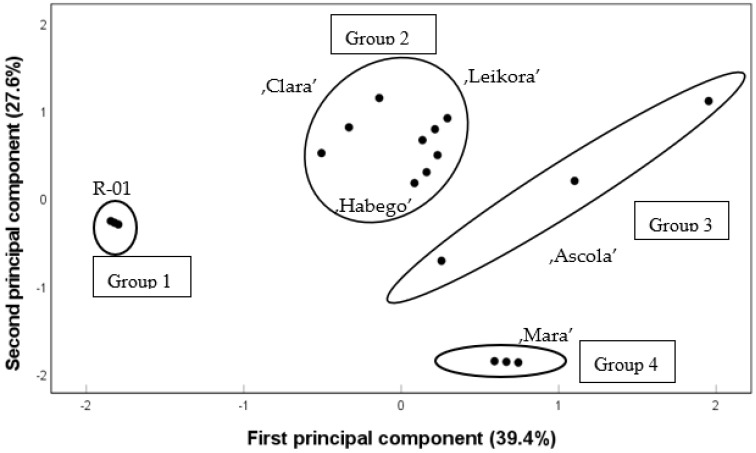
Score plot of PCA for six SB genotypes.

**Figure 2 plants-11-03412-f002:**
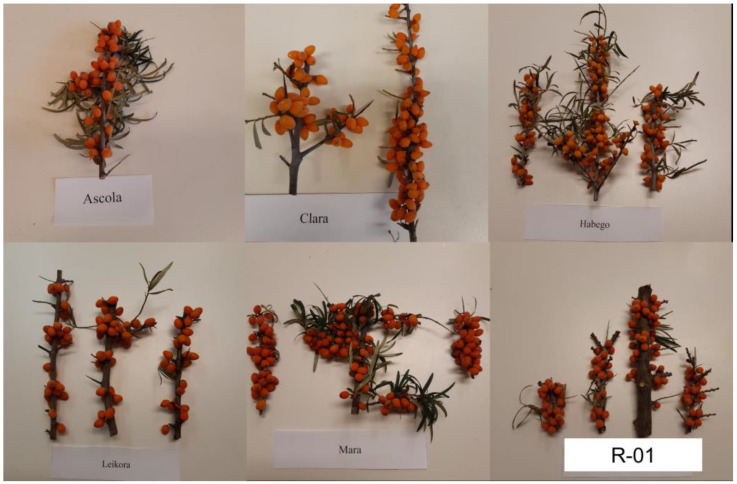
Fruits of the tested SB genotypes.

**Table 1 plants-11-03412-t001:** Size and mass parameters of the berries of SB cultivars.

Cultivars	Height (mm)		Thickness (mm)		Width (mm)		Berry Weight (g)		Seed Weight (g)		Seed/Berry Ratio	
Ascola	9.14 ± 0.42	a	6.96 ± 0.33	a	6.76 ± 0.37	a	0.27 ± 0.05	a	0.027 ± 0.007	b	7.59 ± 1.48	a
Clara	11.15 ± 0.49	b	6.89 ± 0.29	a	6.94 ± 0.28	ab	0.35 ± 0.05	a	0.019 ± 0.001	a	5.52 ± 1.04	a
Habego	10.11 ± 0.45	ab	6.66 ± 0.29	a	6.49 ± 0.29	a	0.25 ± 0.06	a	0.036 ± 0.005	c	6.85 ± 1.13	a
Leikora	12.67 ± 0.47	c	9.17 ± 0.49	b	8.99 ± 0.45	c	0.64 ± 0.07	b	0.029 ± 0.006	b	5.63 ± 1.02	a
Mara	9.88 ± 0.49	a	7.00 ± 0.42	a	6.81 ± 0.35	ab	0.28 ± 0.04	a	0.020 ± 0.002	a	6.61 ± 1.14	a
R-01	9.48 ± 0.39	a	7.72 ± 0.29	a	7.64 ± 0.28	b	0.37 ± 0.04	a	0.032 ± 0.004	b	7.76 ± 1.21	a

The letters indicate significance differences between the varieties. Presented values are means ± SD. (Tukey’s test, *p* < 0.05).

**Table 2 plants-11-03412-t002:** Soluble solid content (SSC) and acid content (TTA) of berries of SB cultivars.

	SSC (°Brix)		TTA (g 100 g^−1^)		Sugar/Acid Ratio	
Ascola	9.1 ± 0.1	ab	2.7 ± 0.1	b	3.4 ± 0.31	a
Clara	7.3 ± 0.2	a	1.9 ± 0.13	ab	3.8 ± 0.39	a
Habego	6.7 ± 0.1	a	2.3 ± 0.29	b	2.9 ± 0.42	a
Leikora	6.3 ± 0.5	a	2.0 ± 0.52	ab	3.1 ± 0.75	a
Mara	10.9 ± 0.6	b	1.4 ± 0.04	a	7.8 ± 0.35	b
R-01	10.2 ± 0.8	b	3.7 ± 0.09	c	2.7 ± 0.27	a

The letters indicate significance differences between the varieties. Presented values are means ± SD. (Tukey’s test, *p* < 0.05).

**Table 3 plants-11-03412-t003:** CIELAB color characteristics of berries of SB cultivars.

	Lightness Value L*		Red-Green Value (a*)		Yellow-Blue Value (b*)		Color Saturation (Chroma)		Hue Angle (Hue)	
Ascola	45.59 ± 0.19	c	0.51 ± 0.02	a	0.58 ± 0.01	a	1.04 ± 0.01	a	48.47 ± 1.56	a
Clara	56.01 ± 0.04	f	14.67 ± 1.46	c	53.26 ± 1.65	d	8.24 ± 0.18	e	74.62 ± 0.99	c
Habego	51.72 ± 0.5	e	14.12 ± 0.2	c	49.68 ± 1.0	d	7.99 ± 0.08	d	74.13 ± 0.11	c
Leikora	50.01 ± 0.8	d	16.89 ± 0.3	c	46.32 ± 1.5	d	7.9 ± 0.11	d	69.96 ± 0.34	bc
Mara	42.84 ± 0.2	b	15.11 ± 0.15	c	25.90 ± 0.4	c	6.4 ± 0.04	c	59.75 ± 0.15	ab
R-01	41.05 ± 0.3	a	7.42 ± 0.5	b	15.89 ± 0.6	b	4.81 ± 0.11	b	65.15 ± 0.81	bc

The letters indicate significance difference between the varieties. Presented values are means ± SD. (Tukey’s test, *p* < 0.05).

**Table 4 plants-11-03412-t004:** Fatty acid composition of SB seeds (mg 100 g^−1^).

	Ascola	Clara	Habego	Leikora	Mara	R-01
Fatty Acid	mg 100 g^−1^	mg 100 g^−1^	mg 100 g^−1^	mg 100 g^−1^	mg 100 g^−1^	mg 100 g^−1^
Saturated fatty acids						
Myristic acid	133.8 ± 36.6	13,055.9 ± 986.5	69.6 ± 5.8	130.2 ± 4.2	88.9 ± 6.0	75.8 ± 0.0
Pentadecanoic acid	69.6 ± 19.7	40.32± 2.9	45.9 ± 3.7	47.05 ± 3.1	53.2 ± 1.6	34.9 ± 0.3
Palmitic acid	11,121.9 ± 2272.9	13,055.9 ± 986.5	13,098.9 ± 144.2	13,246.4 ± 281.8	8472.8 ± 317.7	11,833.7 ± 463.8
Stearic acid	1635.9 ± 348.1	1125.6 ± 87.3	1487.2 ± 16.4	1497.8 ± 28.0	1353.6 ± 45.3	750.5 ± 23.7
Arachidic acid	249.3 ± 55.6	219.2 ± 35.7	276.8 ± 21.8	305.1 ± 8.2	237.5 ± 4.8	137.2 ± 4.3
Behenic acid	75.1 ± 21.6	73.1 ± 14.3	56.5 ± 19.9	93.2 ± 27.2	65.3 ± 2.2	45.4 ± 12.6
Monounsaturated fatty acids						
Palmitoleic acid	8617.0 ± 1736.7	9552.1 ± 684.8	9294.5 ± 110.2	10,661.7 ± 226.4	5543.5 ± 215.5	12,272.7 ± 500.2
Oleic acid	11,611.5 ± 2291.8	11,757.4 ± 808.2	13,567.2 ± 85.6	11,325.6 ± 212.4	10,102.1 ± 432.6	9891.8 ± 366.6
Gondoic acid	20.7 ± 6.3	29.1 ± 0.7	14.9 ± 3.1	18.9 ± 4.2	128.8 ± 2.7	18.7 ± 1.0
Cis-15-tetracosenoic acid	10.5 ± 1.4	6.4 ± 3.2	5.6 ± 7.9		3.1 ± 1.4	7.0 ± 4.5
Polyunsaturated fatty acids						
Linoleic acid	22,571.6 ± 4121.3	13,623.0 ± 261.7	14,756.2 ± 254.7	13,740.7 ± 244.2	20,764.2 ± 1215.2	9332.4 ± 386.6
α-linolenic acid	16,764.2 ± 3338.6	11,963.6 ± 844.8	15,269.2 ± 162.8	13,037.4 ± 270.9	15,570.0 ± 620.6	8436.0 ± 320.2
Dihomo-γ-linolenic acid	16.3 ± 4.0	19.4 ± 0.8	11.4 ± 2.9	12.7 ± 1.6	9.9 ± 1.7	12.7 ± 1.4
ɷ-6/ɷ-3 (linoleic acid/linolenic acid)	1.3	1.13	0.96	1.05	1.33	1.10

Average ± standard deviation.

**Table 5 plants-11-03412-t005:** Fatty acid composition of SB seeds (m/m%).

	Ascola	Clara	Habego	Leikora	Mara	R-01
Fatty Acid	m/m%	m/m%	m/m%	m/m%	m/m%	m/m%
	Saturated fatty acids					
Myristic acid	0.2 ± 0.0	20.8 ± 0.0	0.1 ± 0.0	0.2 ± 0.0	0.1 ± 0.0	0.1 ± 0.0
Pentadecanoic acid	0.1 ± 0.0	0.1 ± 0.0	0.1 ± 0.0	0.1 ± 0.0	0.1 ± 0.0	0.1 ± 0.0
Palmitic acid	15.1 ± 0.3	20.8 ± 0.0	19 ± 0.3	20.2 ± 0.0	13.6 ± 0.1	21.5 ± 0.0
Stearic acid	2.2 ± 0.1	1.8 ± 0.0	2.2 ± 0.0	2.3 ± 0.0	2.2 ± 0.0	1.4 ± 0.0
Arachinic acid	0.3 ± 0.0	0.3 ± 0.0	0.4 ± 0.0	0.5 ± 0.0	0.4 ± 0.0	0.2 ± 0.0
Behenic acid	0.1 ± 0.0	0.1 ± 0.0	0.1 ± 0.0	0.1 ± 0.0	0.1 ± 0.0	0.1 ± 0.0
	Monounsaturated fatty acids					
Palmitoleic acid	11.7 ± 0.2	15.2 ± 0.1	13.5 ± 0.2	16.3 ± 0	8.9 ± 0.1	22.3 ± 0
Oleic acid	15.8 ± 0.2	18.8 ± 0.2	19.7 ± 0.2	17.3 ± 0	16.2 ± 0.0	18.0 ± 0.0
Gondoic acid	0 ± 0	0 ± 0	0 ± 0	0 ± 0	0.2 ± 0.0	0 ± 0.0
Cis-15-tetracosenoic acid	0 ± 0	0 ± 0	0 ± 0	0 ± 0	0 ± 0	0 ± 0
	Polyunsaturated fatty acids					
Linoleic acid	30.7 ± 0.1	21.8 ± 1.3	21.4 ± 0.4	21.0 ± 0	33.2 ± 0.4	17.0 ± 0
α-linolenic acid	22.8 ± 0.3	19.1 ± 0.1	22.2 ± 0.3	19.9 ± 0	24.9 ± 0.1	15.3 ± 0
Dihomo-γ-linolenic acid	0 ± 0	0 ± 0	0 ± 0	0 ± 0	0 ± 0	0 ± 0
	Ratio of unsaturated fatty acids					
Saturated fatty acids (%)	19.0	25.1	23.2	25.5	16.6	27.4
Unsaturated fatty acids (%)	81.0	74.9	76.8	74.5	83.4	72.6
Polyunsaturated fatty acids (%)	53.5	40.9	43.6	40.9	58.1	32.3

Average ± standard deviation.

**Table 6 plants-11-03412-t006:** Correlation values of fatty acids with principal components.

Fatty Acids	PCA1	PCA2
Myristic acid	0.362	0.569
Pentadecanoic acid	0.864	−0.020
Palmitic acid	−0.185	0.943
Stearic acid	−0.627	0.667
Arachinic acid	0.940	0.199
Behenic acid	0.423	0.627
Palmitoleic acid	0.902	−0.299
Oleic acid	0.965	−0.075
Gondoic acid	0.766	0.337
Cis-15-tetracosenoic acid	0.313	−0.823
Linoleic acid	0.065	0.637
*α*-linolenic acid	0.569	0.445
Dihomo-*γ*-linolenic acid	−0.009	0.061

PCA1: first principal component; PCA2: second principal component.

## Data Availability

Not applicable.

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
