# Peer review of "Assessment of Berries of Some Sea Buckthorn Genotypes by Physicochemical Properties and Fatty Acid Content of the Seed"

_plants, 2022, doi:10.3390/plants11243412_

Round 1

Reviewer 1 Report

In general, the manuscript is interesting and should be of interest to plant growers and processors.

I suugest to have it checked by the native English speaker.

The Introduction is well written and gives an appropriate background to the study undertaken. I suggest to replace „19” with the full word „Nineteen” in line 66.

Results section: 

·        Check the values and letters for significance of differences for seed weight in Table 1 as well as the statement in lines 111-112 – „Mara”cultivar was mentioned together with „Leikora” as the largest seed weight (0.02 and 0.029 g, resp.) while in Table 1 „Habego” has a greater weight (0.04 g). All values should have the same accuracy, i.e. the same amount of decimal places after the point.

·        Lines 127-130: The value 10.84 °Brix was reported as the highest while in line 130 the value 10.87 °Brix is given for „Mara”. Which is correct?

·        Table 2: The value for the „Ascola” should be given with the same accuracy as other values in column and completed with SD. I suggest the use of the same decimal accuracy of standard deviation throughout the whole manuscript.

·        The units for acid content in Table 2 (%) and in the text (g 100g-1) should be consistent, i.e. % or g 100g-1 in the whole manuscript.

·        Lines 172-176: Chroma and hue values are calculated on the basis of redness (a*) and yellowness (b*)  values, therefore the statements in those lines should be corrected.

·        Table 5: The value for  linoleic acid for „Leikora” and „R-01”should be given with the same accuracy as other values in that table.

·        Lines 213-214: The content of linolenic acid in other studies is reported at the level 20-35%, in own study 19.1-24.9% and as being higher that in those other studies. I can not agree with that statement.

·        Line 29 (in second part of the manuscript): I suggest to correct the sentence ”All out investigated varieties exceed these values, by 8.9-22.3%” to „All out investigated varieties exceed these values, and were in the range 8.9-22.3%”.

·        Line 35 (in second part of the manuscript):I suggest to replace „…it occurs more amount …” with „ …it occurs in higher amount ……., and lower in the seed”.

Materials and Methods section:

·        Lines 63-64: Explain, how the stone/seed ratio has been calculated, please.

·        All laboratory equipment used in the study should be identified by the type accompanied by the name, town and country of the manufacturer. The same applies to the statistical analysis software.

Author Response

Dear Reviewer,

thank you for your comments on the manuscript, with which we agree. You can find the corrections in the attachment.

Kind regards, Gitta Ficzek

Reviewer 2 Report

Máté et al.'s research  on "Comparative Study of Physicochemical Properties and Seed Oils of some Sea Buchthorn Cultivars and one Hungarian Candidate in Aspect of Processability" is fascinating. However, revision needs to be done to improve the articles. I have incorporated my comments into the attached file. Besides, I have corrected the grammatical errors and written them in American English (red lines). I am willing to review it again after revision. Thanks

Author Response

(The authors gave the same response as above.)

Round 2

Reviewer 2 Report

Thanks for the revision. 

This is a significant improvement from the previous manuscript. However, I have a few comments that authors need to incorporate into their manuscript before it is accepted for publication. 

Comment 1

Figure 2

What do groups 1, 2, - mean? Explain them in Figure 2.

Comment 2

3.4. Determination of fatty acid profile by GC-FID method

How much of the lyophilized sample (1 mg, 2mg….)

Give a detailed procedure for the extraction. Write it so that someone who wants to reproduce the method will find it easy. 

Comment 3

In FID, what was the program mode for the FID? 

Give a detailed procedure for the extraction. Write it so that someone who wants to reproduce the method will find it easy. 

The FID program should be like this 

For example 

short-time programmed heating-up method. The initial temperature was …. maintaining for min, raised to ….at a rate of  …..°C/min, then to °C at a rate of °C/min, maintaining for .. min, and finally, the post-run was kept at °C for 1 min. 

For a long-time programmed heating-up

method, the initial temperature was … °C maintaining for …. min, raised to … °C at a rate of …°C/min, then to … °C  at a rate of .. °C/min maintaining for .. min, and finally, the post-run was kept at … °C for .. min. 

Comment 4

add more discussion in the principal component analysis. What are the relationships between various eigenvalues? From the eigenvalues, which one significantly contributed to the PCA? Also, add the eigenvector tables in the supplementary results. 

Comment 5

Since the authors looked at various genotypes of Sea Buckthorn berries, it will be better to include pictures of the various cultivars used since you calculated the color parameters. That will help improve the manuscript. 

Author Response

Dear Reviewer,

thank you for your reviewsion again, we have supplemented the manuscript based on your suggestions.
We hope you will accept the changes we have corrected.

Kind regards,
Gitta Ficzek
